# Smooth muscle electromyography for detecting major alterations in the estrus cycle in rats

**Kalman F. Szucs[iD], Dora Vigh, Mohsen Mirdamadi, Reza Samavati, Tamara Barna[iD], Annamaria Schaffer, Karmen Alasaad, Robert Gaspar[iD]***

Department of Pharmacology and Pharmacotherapy, Albert Szent-Györgyi Medical School, University of Szeged, Szeged, Hungary

* gaspar.robert@med.u-szeged.hu

**Data Availability Statement:** All relevant data are within the manuscript and its Supporting Information files.

## Abstract

Determining the female animal cycle is crucial in preclinical studies and animal husbandry. Changes in hormone levels during the cycle affect physiological responses, including altered contractility of the visceral smooth muscle. The study aimed to identify estrus and anestrus using smooth muscle electromyographic (SMEMG) measurements, in vivo fluorescent imaging (IVIS) and in vitro organ contractility of the uterus and cecum. The study involved sexually mature female *Sprague-Dawley* rats, aged 10–12 weeks. The rats received a daily injection of cetrorelix acetate solution for 7 days, while another group served as the control. The animals were subjected to gastrointestinal and myometrial SMEMG. The change in αvβ3 integrin activity was measured with IVIS in the abdominal cavity. Contractility studies were performed in isolated organ baths using dissected uterus and cecum samples. Plasma samples were collected for hormone level measurements. A 3-fold increase in spontaneous contraction activity was detected in SMEMG measurements, while a significant decrease in αvβ3 integrin was measured in the IVIS imaging procedure. Cetrorelix reduced the level of LH and the progesterone / estradiol ratio, increased the spontaneous activity of the cecum rings, and enhanced KCl-evoked contractions in the uterus. We found a significant change in the rate of SMEMG signals, indicating simultaneous increases in the contraction of the cecum and the non-pregnant uterus, as evidenced by isolated organ bath results. Fluorescence imaging showed high levels of uterine αvβ3 integrin during the proestrus-estrus phase, but inhibiting the sexual cycle reduced fluorescence activity. Based on the results, the SMEMG and IVIS imaging methods are suitable for detecting estrus phase alterations in rats.

## Introduction

Determining the cycle of female animals is a critical point in preclinical studies and even in animal husbandry. In rats, the levels of follicle stimulating hormone (FSH) and luteinizing hormone (LH) begin to rise with puberty, indicating the end of the anestrus phase [1]. The

**Funding:** National Research, Development and Innovation Fund, Hungary (2018-1.3.1-VKE-2018-00014 project) Project No. TKP2021-EGA-32 was implemented with the support provided by the Ministry of Innovation and Technology of Hungary Szegedi Tudományegyetem, Open Access Fund 6917, Dr Robert Gaspar.

**Competing interests:** The authors have declared that no competing interests exist.

cycle of sexually mature rats lasts on average 4–5 days and is characterized as: proestrus, estrus, metestrus and diestrus, which can be determined according to the cell types observed in the vaginal smear [2]. LH levels rise suddenly at the end of the proestrus stage. This surge causes the maturation of the ovarian follicles, which is followed 10–12 hours later by follicular rupture and ovulation at the beginning of the estrous phase. Accurate tracking of the sex cycle allows the selection of appropriate animals for experiments or even to achieve timed mating [3, 4].

The physiological processes induced by the increase in the LH level can be inhibited by drugs. Cetrorelix is a potent and selective competitive antagonist of the GnRH receptor. Reversibly reduces LH and FSH from the anterior pituitary gland, thus preventing ovulation and inhibiting the production of sex hormones E2 and P4 sex hormones [5].

Changes in hormone levels during the cycle greatly influence physiological responses [6] including altered contractility of the visceral smooth muscle. The effect of sex hormones on uterine contraction has long been known: progesterone (P4), which is responsible for maintaining pregnancy, significantly reduces uterine activity, while contractions become more frequent and stronger before birth, when the plasma estrogen/P4 ratio increases [7].

In addition to the reproductive functions of P4 and estrogen, they exert functions in the entire gastrointestinal tract (GI), which is strongly correlated with the highest plasma concentrations of gonadotropic hormones. P4 inhibits intestinal smooth muscle cell contraction, in part by increasing nitric oxide synthesis and in part by inhibiting Rho kinase, which together induce smooth muscle relaxation [8–10]. These changes can be identified by measuring myoelectric slow waves. The myoelectric activity of the GI tract [11] and the uterus [12], and therefore the extent of contractions, can be well measured through the abdominal wall in vivo using smooth muscle electromyography (SMEMG). The method continuously monitors the myoelectric signals of the visceral smooth muscle of rats by frequency filtration after a fast Fourier transformation.

αvβ3 integrin, belonging to the family of transmembrane cell surface proteins, plays an important role during angiogenesis and tissue neovascularization [13], therefore, it could be one of the new targets of antitumor therapy in the last decade. As it is expressed in large quantities in activated endothelial cells, it can be used as an antitumor target and for in vivo imaging in the case of the most common carcinomas and metastases (glioblastoma, melanomas, ovarian, breast, and prostate) [14, 15]. However, it is known that in the proestrus-estrus phase, neovascularization and mucosal proliferation increase to a great extent [2], in which αvβ3 integrin may also play an important role [16].

The aim of our study was to identify physiological estrus and cetrorelix-induced anestrus in rats using SMEMG measurements. We paralleled the changes with an in vivo fluorescent imaging system (IVIS) detecting αvβ3 integrin, an estrus cycle impedance monitor [17] and in vitro isolated organ contractility [18] of the uterus and cecum.

## Materials and methods

### Housing and handling of the animals

The study was carried out using sexually mature female *Sprague-Dawley* rats (10–12 weeks old, body weight 160–200 g). The animals were treated in accordance with the European Communities Council Directive (2010/63/EU) and the Hungarian Act for the Protection of Animals in Research (Article 32 of Act XXVIII). All experiments involving animal subjects were carried out with the approval of the Hungarian Ethical Committee for Animal Research (registration number: XIII./72/2020).

*Sprague-Dawley* rats (Animalab Ltd., Vác, Hungary) were housed at $22 \pm 3°C$ and a relative humidity of 30–70%, in a 12 h light/12 h dark cycle. Standard rodent pellet food (Animalab Hungary Ltd.) and tap water were provided *ad libitum*.

The estrus cycle of healthy young adult female rats was followed and measured daily using the Estrus Cycle Monitor (IM-01, MSB-MET Ltd., Balatonfüred, Hungary). Rats with vaginal impedance values of 4.5 to 7.5 kΩ were considered to be in the proestrus phase and were included in the experiment.

Animals in the drug-treated group (n = 10) received a daily intraperitoneal injection of 0.5 mg/bwkg cetrorelix acetate solution (Sigma-Aldrich Ltd., Budapest, Hungary) for 7 days [19, 20], and the resulting changes were compared with those of the control group of rats (n = 9).

## Smooth muscle electromyographic (SMEMG) measurements

Two hours before and during the detection of myoelectric signals, food and water were withdrawn. Rats were anesthetized with isoflurane inhalation, then a pair of bipolar disk electrodes was fixed subcutaneously 1 cm below the midline above the abdominal cavity. The SMEMG activity was detected for 30 min by an online computer and amplifier system with the S.P.E.L. Advanced ISOSYS Data Acquisition System (MSB-MET Ltd., Balatonfüred, Hungary).

The SMEMG records from both groups of rats were filtered for a frequency of 1–3 cpm and analyzed by fast Fourier transformation [11, 12], then the maximum power spectrum density ($PsD_{max}$) values for 30-min periods were statistically evaluated.

## In vivo imaging protocol

Rats were injected with a potent αvβ3 integrin-selective imaging agent; IntegriSense$^{TM}$ 680 (PerkinElmer Ltd., Boston, USA) diluted in phosphate buffered saline (PBS), intravenously into the tail vein. An abdominal incision was made to allow live imaging under isoflurane anesthesia. Imaging was performed 24 hours after injection with the IVIS Lumina LT III system (PerkinElmer Ltd., Waltham, USA) with 675 nm excitation and Cy5.5 emission filters, auto exposure time, and a binning factor of 2. The fluorescence intensity was represented by a multicolor scale ranging from red (less intense) to yellow (more intense). Fluorescence was normalized by uterus size and expressed as the average radiant efficiency ([photons/s/cm$^2$/steradian]/[μW/cm$^2$]) of selected regions of interest (ROIs).

## Plasma sample collection and hormone analysis

At the end of the IVIS measurements, 3 ml of blood samples were collected by cardiac puncture and divided into 1-ml tubes containing $K_3EDTA$ (0.6 mg/tube) and centrifuged (1700 ×g, 10 min, 4°C) to separate plasma. Plasma samples were stored at -20°C until the hormone test.

Plasma concentrations of LH, P4, and estradiol (E2) were measured by enzyme-linked immunosorbent assay (ELISA, from Wuhan Fine Biotech Co., Ltd.), according to the manufacturer's manual.

## Contractility studies in isolated organ bath

Animals were sacrificed by cardiac puncture under isoflurane anesthesia. The uterus and cecum were dissected from non-pregnant rats, cleaned of fat and connective tissues, and rinsed with de Jongh solution (composition: 137 mM NaCl, 3 mM KCl, 1 mM $CaCl_2$, 1 mM $MgCl_2$, 12 mM $NaHCO_3$, 4 mM $NaH_2PO_4$, 6 mM glucose, pH = 7.4) for uterus and Tyrode solution (composition in mM: 137 NaCl, 3 KCl, 1 $CaCl_2$, 1 $MgCl_2$, 12 $NaHCO_3$, 0.4 $NaH_2PO_4$, 6 glucose, pH 7.4) for cecum [18, 21].

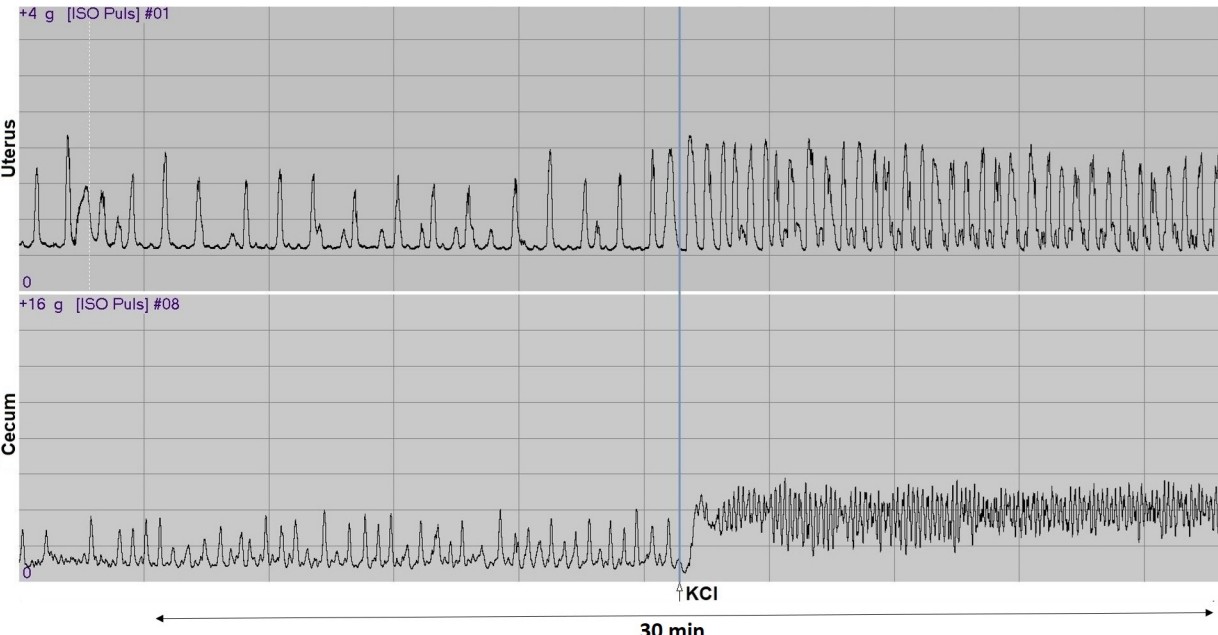

**Fig 1. Representative 30-min records of the isolated organ bath study.** Spontaneous rhythmic contractions were enhanced in the rings of the uterus (upper channel) and cecum (lower channel) by KCl (25mM) prepared from rats.

The uterus and cecum samples were cut into 5-mm-long muscle rings, then these cleaned muscle strips were individually mounted on tissue holders and immediately placed in the isolated organ bath chambers. Each chamber contained 10 ml of appropriate buffer, and the temperature was maintained at 37°C with continuous carbogen (95% $O_2$ + 5% $CO_2$) support. After mounting, the initial resting tension was set at 1.5 g and the strips were allowed to equilibrate for 60 min with a buffer change every 15 min.

After the incubation period, spontaneous control contractions of the smooth muscle were evoked by KCl (25 mM). The activity of the tissue rings was measured with a gauge transducer (SG-02, MSB-MET Ltd., Balatonfüred, Hungary). For recording and analysis, we used the SPEL Advanced ISOSYS Data Acquisition System (MSB-MET Ltd., Balatonfüred, Hungary) (Fig 1). Areas under curves (AUC) of 5-minute periods were evaluated, and the effects of KCl were expressed as a percentage of spontaneous contractions [18].

## Statistical analysis

Sex hormone levels, AUC, average radiant efficiency and $PsD_{max}$ values were determined and compared using an unpaired t-test. The p values of the unpaired t-tests indicating statistically significant differences are shown in the respective figures. Statistical analyzes were performed using the statistical program Prism 10.1. (GraphPad Software LLC, RRID:SCR_002798), and the level of statistical significance was set at $p < 0.05$.

## Results

On the day of inclusion in the study of rats, the vaginal impedance values were typical for the proestrus phase. The 7-day cetrorelix treatment significantly reduced vaginal impedance compared to the initial values (Fig 2).

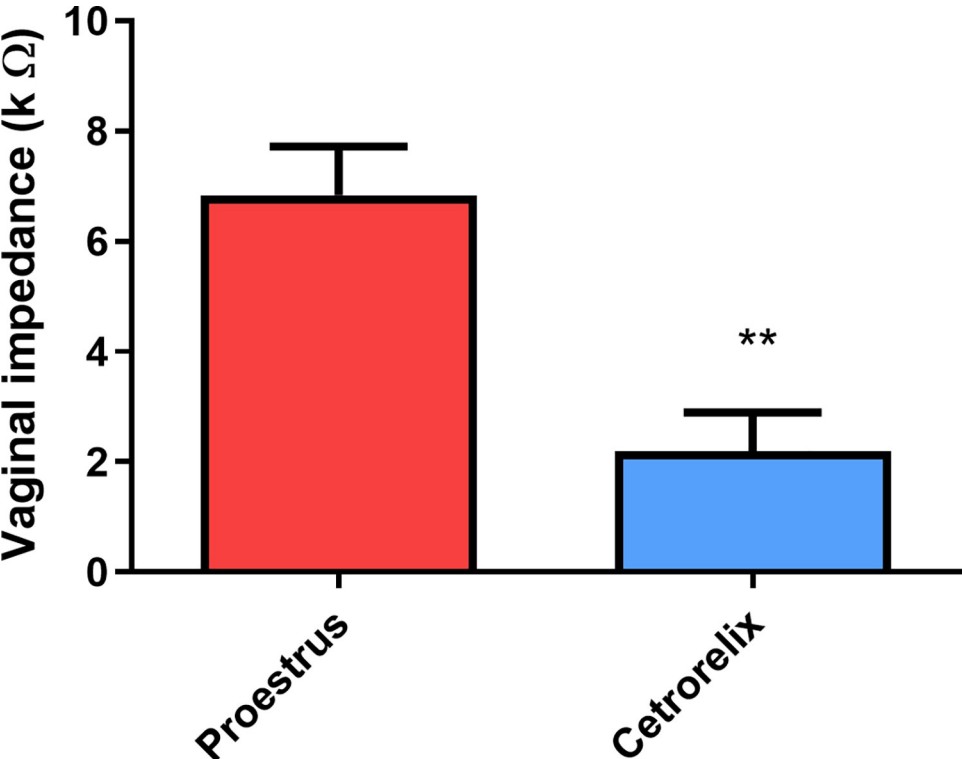

**Fig 2. Vaginal impedance (± SEM) values measured with the estrus cycle monitor.** A significant decrease in values was obtained due to the 7-day cetrorelix treatment. (\*\*: $p < 0.01$).

### In vivo smooth muscle electromyography

Smooth muscle myoelectric activity was detected in anesthetized rats and expressed as $PsD_{max}$ in the frequency range of 1–3 cpm (Fig 3).

The rate of increase in the $PsD_{max}$ value was approximately three times in the group treated with cetrorelix compared to the value of the control group. This change in signal intensity indicates a significant increase in spontaneous uterine or cecal contractions (Fig 4).

### In vivo imaging

The intensity of αvβ3 integrin staining was measured in the IVIS imaging procedure. The fluorescent probe showed an increase in intensity in non-pregnant uterine horns compared to background radiation, indicating an increased αvβ3 integrin activity. However, the treatment with cetrorelix significantly reduced the fluorescent intensity of the uterus (Fig 5).

The effect of cetrorelix treatment was quantified after evaluating the ROIs of the images. Treatment reduced the intensity of αvβ3 integrin staining to less than half in the non-pregnant uterus compared to the control group (Fig 6).

### Enzyme-linked immunosorbent assay

The 7-day cetrorelix treatment significantly reduced LH and P4 levels compared to the control group. The level of E2 was already low in the control group, so the treatment did not cause any further changes; however, the ratio of P4/E2 hormones was also significantly reduced by cetrorelix treatment (Fig 7).

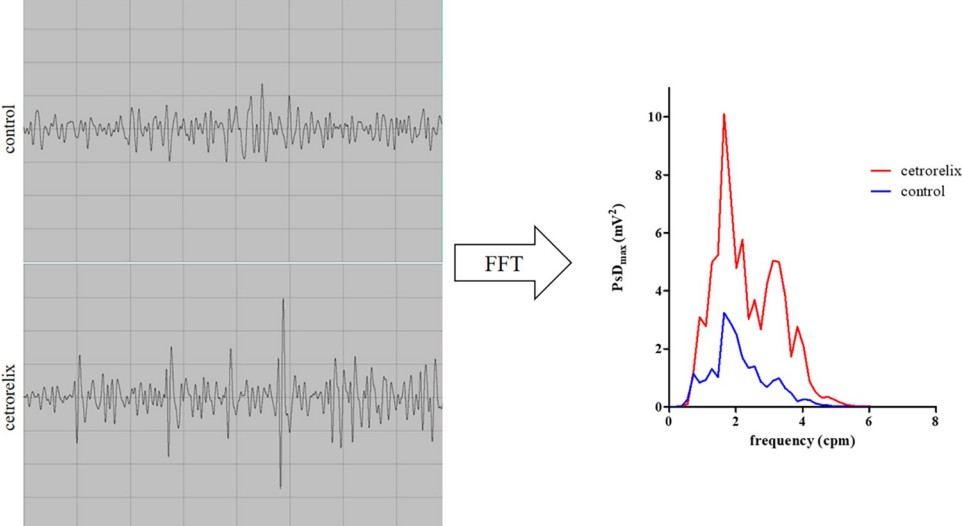

**Fig 3. Representative primary myoelectric signals of 30-min records from the SMEMG study.** Myoelectric signals from the myometrium were detected with silver disk electrodes in the control group (upper channel), and in the cetrorelix-treated group (lower channel). The recorded myoelectric signals were then analyzed by fast Fourier transformation (FFT). Changes in $PsD_{max}$ reflect changes in cecal or uterine smooth muscle contractions.

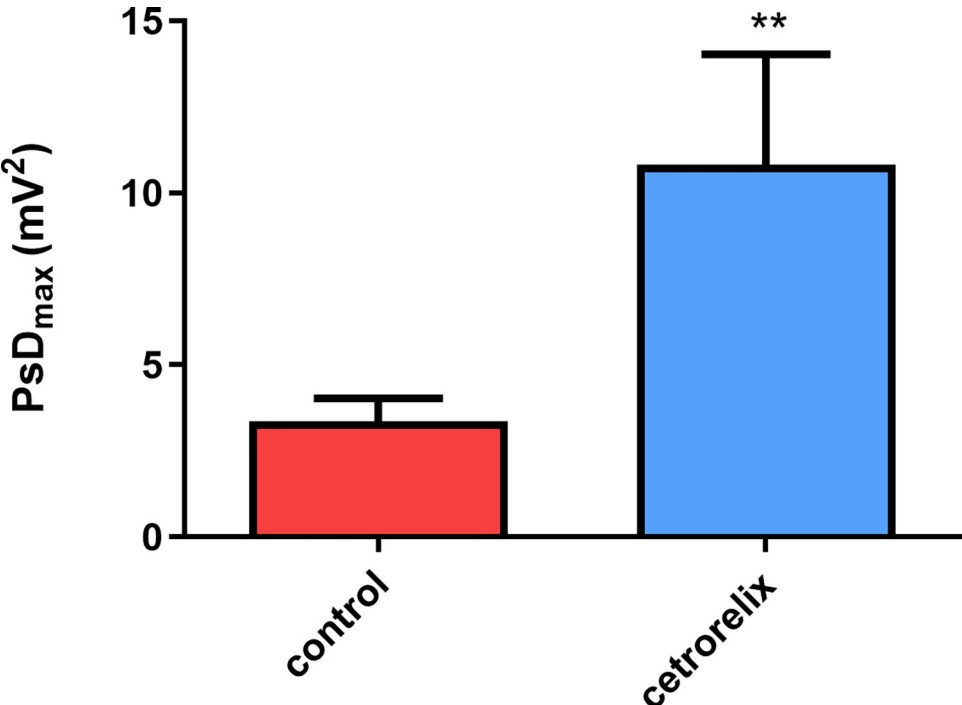

**Fig 4. Changes in the power spectrum density maximum values (PsDmax) after cetrorelix treatment in anesthetized rats detected by in vivo smooth muscle electromyography.** Values are expressed as $PsD_{max} \pm SEM$, showing muscle contraction strength. A significant increase in the activity of spontaneous contractions was detected with cetrorelix treatment. (\*\*: $p < 0.01$).

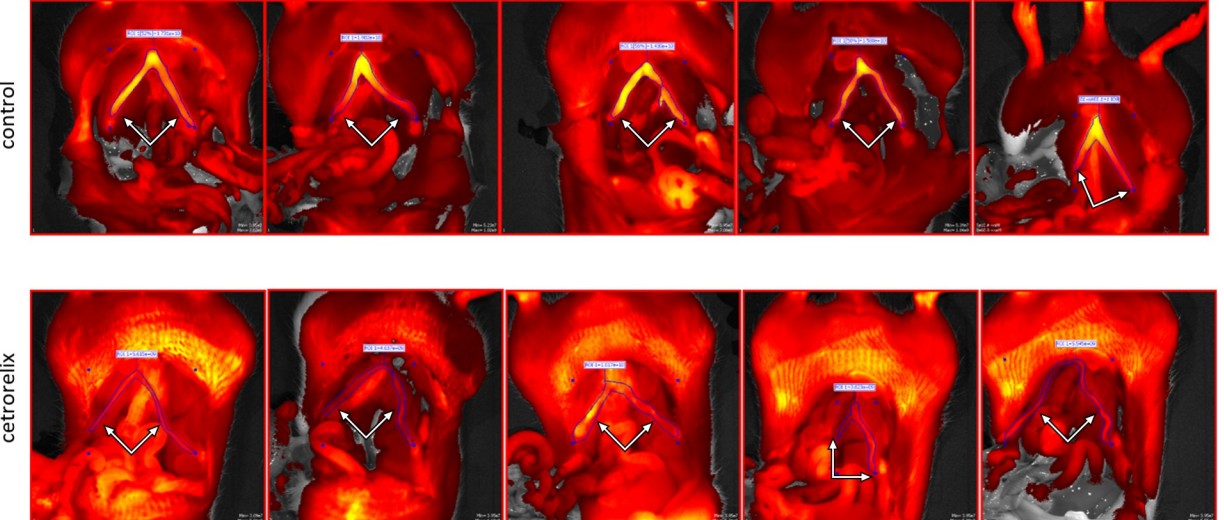

**Fig 5. Representative fluorescent records of IntegriSense 680® dye-labeled αvβ3 integrin in control and cetrorelix-treated non-pregnant rats.** The yellowish intensity staining seen in the uterus indicates αvβ3 integrin, which is much brighter during the fertilization period. Cetrorelix treatment reduced αvβ3 integrin expression, which caused a significant decrease in intensity (bottom image row). The white arrows indicate the uterine horns.

## In vitro contractility

Smooth muscle contractions were detected in our isolated organ bath study. The spontaneous activity of the cecum rings was significantly increased after cetrorelix treatment compared to the AUC value in the control group. However, there was no significant change in uterine activity (Fig 8).

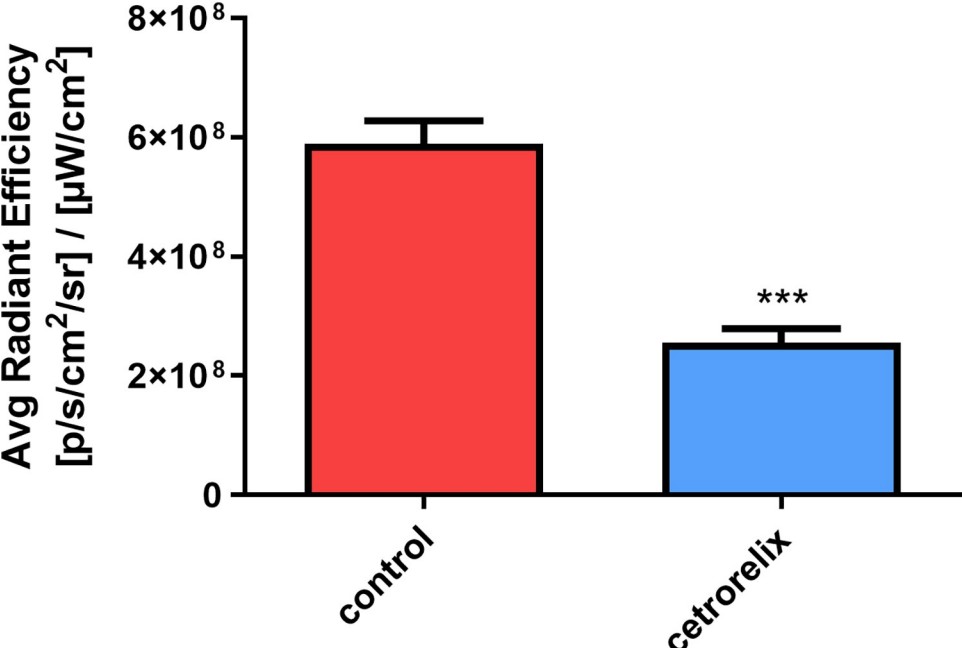

**Fig 6. Average radiant efficiency (± SEM) in equally sized regions of interest (ROIs) of the non-pregnant uterus.** A significant decrease in αvβ3 integrin was observed in the PET imaging procedure. (***: $p < 0.001$).

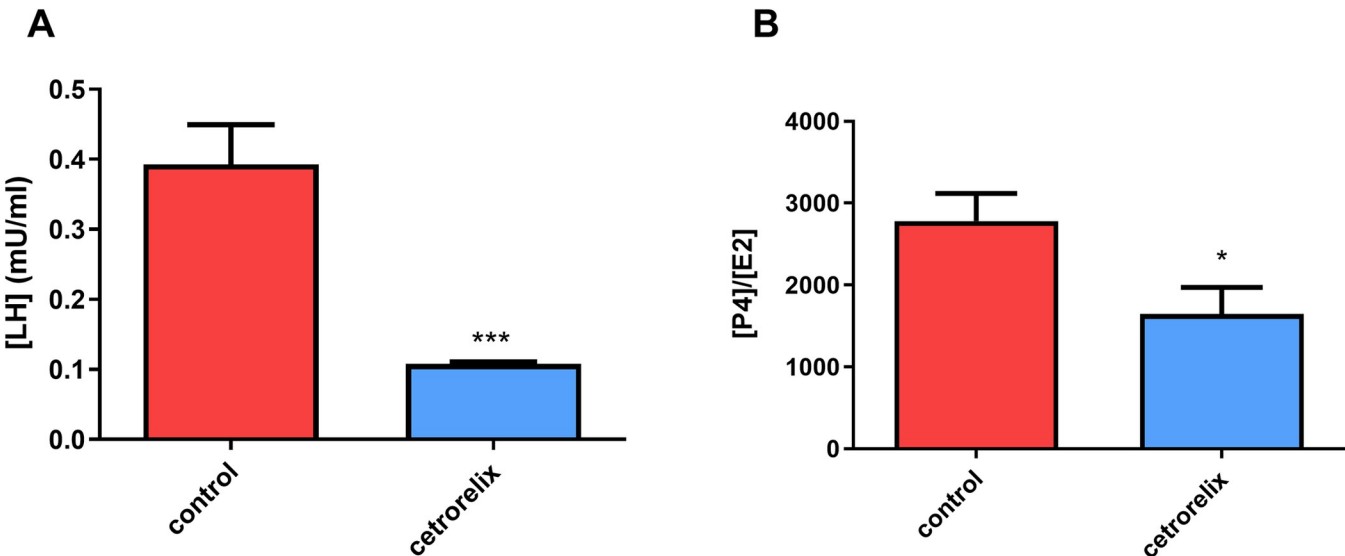

**Fig 7.** Alterations in the plasma (A) luteinizing hormone (LH) and (B) sex hormone ratio (P4/E2) as a result of cetrorelix treatment (red columns: estrus phase, blue columns: anestrus phase). The LH level was significantly decreased by cetrorelix compared to the control group. Treatment with cetrorelix also significantly reduced the value of P4/E2. (*: $p < 0.05$; ***: $p < 0.001$).

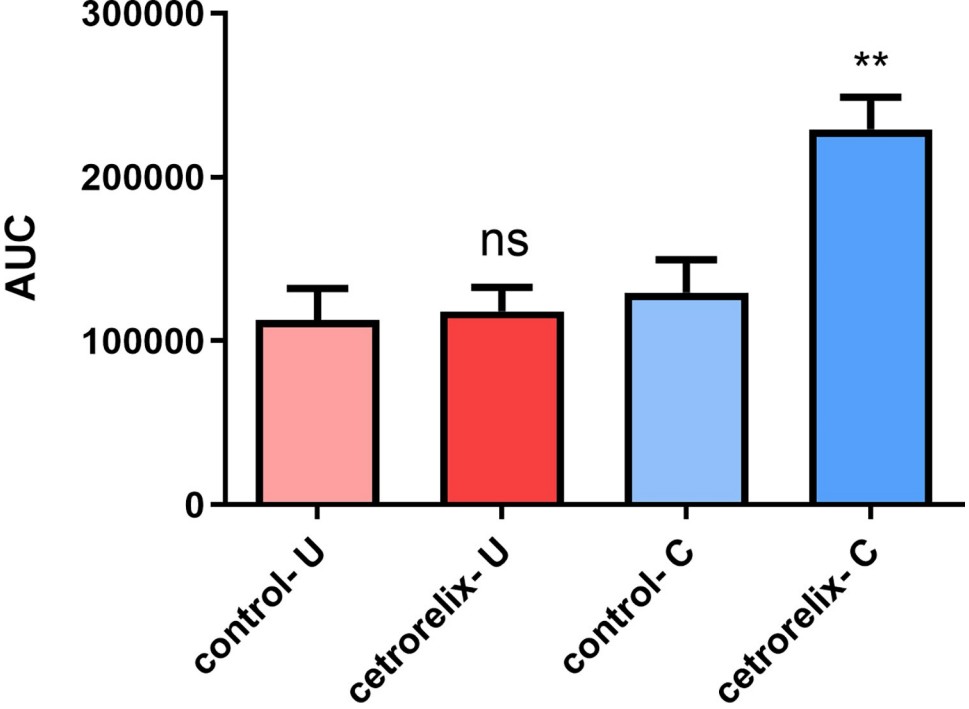

**Fig 8. Effects of cetrorelix treatment on changes in spontaneous contractions of the non-pregnant uterus (U) and the cecum (C).** The change in contraction was evaluated through the area under the curve (AUC) ± SEM. No significant effect was observed in the investigated myometrium samples; in contrast, a significant increase was obtained for the cecum. (U: uterus; C: cecum; ns: non-significant; **: $p < 0.01$).

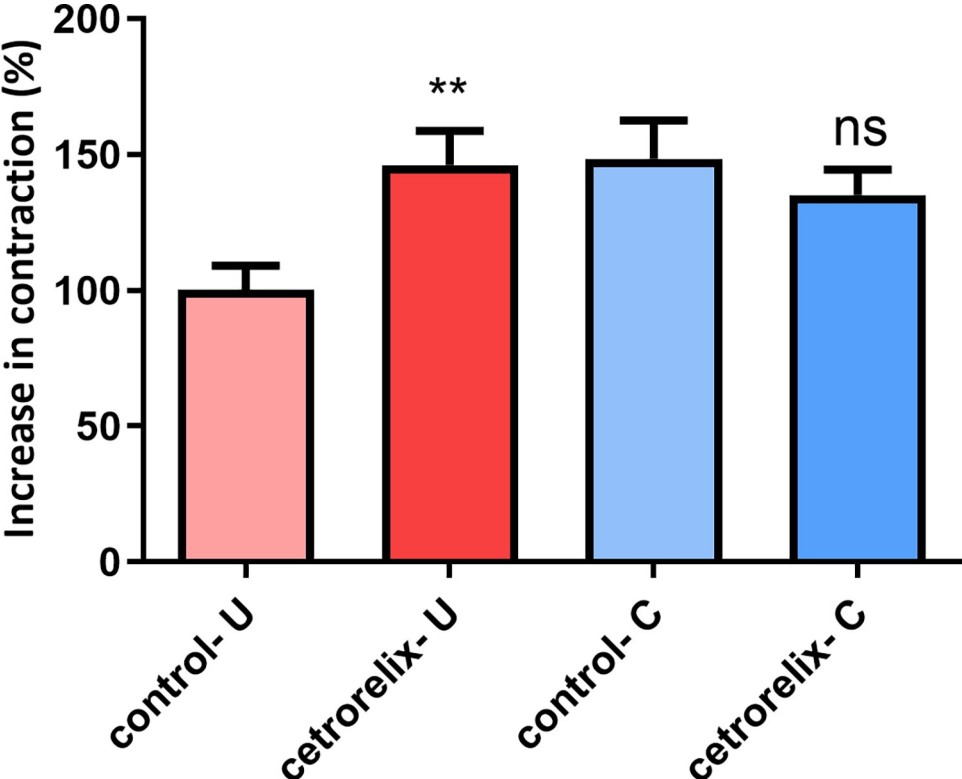

**Fig 9. Effects of cetrorelix treatment on changes in the KCl-evoked contractions of the non-pregnant uterus (U) and the cecum (C).** The relative change in contractions was obtained in each case compared to spontaneous contractions and was expressed as % ± SEM. A significant effect was observed on the investigated uterus samples. (U: uterus; C: cecum; ns: non-significant; **: $p < 0.01$).

KCl-elicited contractions were also examined in non-pregnant uterus and cecum. In contrast to the spontaneous response, cetrorelix treatment significantly increased KCl-evoked contractions in the uterus, while it did not modify cecal activity (Fig 9).

## Discussion

SMEMG, as a non-invasive measurement method, is increasingly used for research and diagnostic purposes. The focus of research to date has been mainly on GI motility studies, animal breeding [22] and preclinical-clinical translational studies [23–25] have also been published on the topic, confirming the usability of the technology; furthermore, with the validation of the method, a human diagnostic tool is now available [26]. A previous study revealed that uterine and colon signals are difficult to separate with the use of in vivo SMEMG measurement because the frequency range of signals obtained after FFT analysis is between 1–3 cpm for both organs, and in addition, the muscle mass of the non-pregnant uterus is significantly lower than that of the colon [12].

In our current study, treatment with the GnRH antagonist cetrorelix increased the contractions of the visceral smooth muscle, i.e. the cecum and uterus. With the SMEMG method, we detected enhanced spontaneous abdominal smooth muscle contractions. The results of the in vivo measurement were confirmed in isolated organ bath studies. Treatment with cetrorelix significantly increased the strength of spontaneous contractions in the cecum. On the contrary, KCl increased the contractions of isolated tissue samples compared to spontaneous

activity in both groups, but a further increase was observed only in the cetrorelix-treated non-pregnant uterus.

In our fluorescence imaging studies, we showed that the uterine level of αvβ3 integrin is high in rats during the proestrus-estrus phase, which is closely related to the vascularization of the uterus [2]. However, when we inhibited the estrus cycle of the animals, thus mucosal proliferation, the fluorescence activity of αvβ3 integrin decreased dramatically.

The well-known hormonal effect of cetrorelix is behind the detected changes. GnRH antagonists have been used for years in the treatment of infertility, in assisted reproductive technology [27], and even in the treatment of leiomyoma or endometriosis [28]. Furthermore, low-dose cetrorelix treatment was also shown to reduce the symptoms of benign prostatic hyperplasia in male rats. This is manifested through an indirect decrease in sex hormone levels and a direct reduction in inflammatory cytokine levels [29]. This effect is achieved through its inhibitory effect on the hypothalamic-pituitary-ovarian axis. Although a physiological decrease in P4 and E2 levels is expected due to the decrease in LH levels, this would not justify such an increase in smooth muscle contraction. However, contractility is more accurately determined by the relative amounts of P4 and E2, rather than by changes in individual hormone levels [30]. Due to these hormonal changes, the routinely applied vaginal impedance measurement also confirmed the effect of cetrorelix, in agreement with our newly applied methods.

Repetitive treatment with cetrorelix stopped the estrus cycle of rats, thus decreasing LH levels and the P4 / E2 ratio; therefore, the rats entered the anestrus phase. This large-scale change in sex hormone levels expressed contrasting differences between estrus and anestrus and had a detectable effect on uterine and even GI contractions. We were able to detect this increase in contraction with SMEMG in awake rats. Furthermore, by using the fluorescent IVIS method, accurate results can be obtained regarding cell proliferation in the uterus of rats between the two endpoints of their estrus cycle.

A further goal with our method would be to detect subtle changes, such as the successive stages of the estrus cycle, even together with the measurement of the αvβ3 integrin level, with high accuracy. The SMEMG method might also be suitable for the complex measurement of reproductive processes in other mammals or in female patients or healthy volunteers after clinical translation.

## Supporting information

**S1 Dataset. Vaginal impedance values measured with the estrus cycle monitor before and after 7 days of cetrorelix treatment (Fig 2).**
(XLSX)

**S2 Dataset. Maximum values of power spectrum density in 2 different groups of rats detected by in vivo smooth muscle electromyography (Fig 4).** PsD: power spectrum density.
(XLSX)

**S3 Dataset. The average radiant efficiency values in equally sized regions of interest of the non-pregnant uterus in the control and cetrorelix treated rats (Fig 6).**
(XLSX)

**S4 Dataset. Changes in plasma luteinizing hormone levels and P4/E2 sex hormone ratios as a result of cetrorelix treatment (Fig 7).** LH: luteinizing hormone, P4: progesterone, E2: estradiol.
(XLSX)

**S5 Dataset. Differences in spontaneous contractions of non-pregnant uterus and cecum samples in the control and cetrorelix treated rats ([Fig 8]).** The changes in contraction were evaluated by the area under the curve. AUC: area under the curve.
(XLSX)

**S6 Dataset. The changes in the KCl-evoked contractions of the non-pregnant uterus and the cecum in the control and cetrorelix treated rats ([Fig 9]).** The relative change in contractions was obtained in each case compared to spontaneous contractions and was expressed as %.
(XLSX)

## Acknowledgments

The technical support of Zoltánné Csiszár is greatly appreciated.

## Author Contributions

**Conceptualization:** Kalman F. Szucs, Robert Gaspar.

**Formal analysis:** Kalman F. Szucs, Tamara Barna, Annamaria Schaffer.

**Funding acquisition:** Robert Gaspar.

**Investigation:** Kalman F. Szucs, Dora Vigh, Mohsen Mirdamadi, Reza Samavati, Tamara Barna, Annamaria Schaffer, Karmen Alasaad.

**Methodology:** Kalman F. Szucs, Dora Vigh, Mohsen Mirdamadi, Reza Samavati, Tamara Barna, Annamaria Schaffer, Robert Gaspar.

**Resources:** Robert Gaspar.

**Writing – original draft:** Kalman F. Szucs.

**Writing – review & editing:** Robert Gaspar.

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
