## [Decision Letter · Decision Letter 0]

24 Jun 2024

PONE-D-24-11044Smooth muscle electromyography for detecting major alterations in the estrus cycle in ratsPLOS ONE

Dear Dr. Gaspar,

Thank you for submitting your manuscript to PLOS ONE. After careful consideration, we feel that it has merit but does not fully meet PLOS ONE’s publication criteria as it currently stands. Therefore, we invite you to submit a revised version of the manuscript that addresses the points raised during the review process.

We look forward to receiving your revised manuscript.

Kind regards,

Abeer El Wakil, PhD

Academic Editor

PLOS ONE

Journal Requirements:

"National Research, Development and Innovation Fund, Hungary (2018-1.3.1-VKE-2018-00014 project)

Project No. TKP2021-EGA-32 was implemented with the support provided by the Ministry of Innovation and Technology of Hungary "

3. In the online submission form, you indicated that [Data are available for request]. 

Reviewers' comments:

Reviewer's Responses to Questions

**Comments to the Author**

1. Is the manuscript technically sound, and do the data support the conclusions?

Reviewer #1: Yes

Reviewer #2: Yes

Reviewer #3: Yes

2. Has the statistical analysis been performed appropriately and rigorously? 

Reviewer #1: Yes

Reviewer #2: Yes

Reviewer #3: Yes

3. Have the authors made all data underlying the findings in their manuscript fully available?

Reviewer #1: Yes

Reviewer #2: Yes

Reviewer #3: Yes

4. Is the manuscript presented in an intelligible fashion and written in standard English?

Reviewer #1: Yes

Reviewer #2: Yes

Reviewer #3: Yes

5. Review Comments to the Author

Reviewer #1: Some Remarks to the Authors:

1) It would be better to give a brief and appropriate explanation about Cetrorelix and its pharmacological mechanism of action in the Introduction section.

2) In the Methods section, which variance analysis (ANOVA, etc) was performed for multiple comparisons should be stated in the Statistical analysis sub-section.

3) References should be added to the use of Jongh's solution for the uterus, Tyrode's solution for the cecum, and the contractility study method.

4) For the Figure 1: The figure should be adjusted correctly, as if KCl application on uterine contractions begins at the 15th minute rather than the 16th minute.

Thanks for your work.

Reviewer #2: Comments for the authors:

In this paper, the authors investigated smooth muscle electromyography for detecting major alterations in the estrus cycle in rats. This study is well-designed, the language is easy to understand, the style is enjoyable, and it contains scientific novelties. Their research results are innovative, as they have both scientific and health relevance and may have economic (e.g., animal husbandry) implications.

Minor Remarks:

1. What is the basis for the dose and duration of cetrorelix acetate treatment?

2. What practical indications are that de Joung solution was used for uterus samples and Tyrode buffer for cecum samples in isolated organ baths?

3. Line 153 of the results section lacks a precise description of the in vivo smooth muscle electromyography result. Supplement the legend of Figure 3 with the Fourier transform FFT notation!

4. I think the bibliographic reference in line 158 (the legend of Figure 3) is redundant. Literature 11 can be found in the methods section and the summary also.

5. In line 175: "Cetrorelix suppressed staining in the uterus (bottom image row)." I suggest instead, for the reader's ease of understanding, "Cetrorelix treatment reduced αvβ3 integrin expression, which caused a significant decrease in intensity."

6. In Figure 5, I propose to mark the uterine horns with a white arrow in each image.

7. Why the effect on KCl-induced contractions was considered to be justified in addition to the spontaneous uterine contractility study.

Reviewer #3: The authors have found simultaneous increases in the contraction of the cecum and the non-pregnant uterus when they used smooth muscle electromyographic (SMEMG) in the female rats. They have also observed that fluorescence imaging showed high levels of uterine αvβ3 integrin during the proestrus-estrus phase, but inhibiting the sexual cycle reduced fluorescence activity. Based on the results, the authors have suggested that the SMEMG and IVIS imaging methods are suitable for detecting estrus phase alterations in rats.

As the authors mentioned, the SMEMG method might be suitable for the complex measurement of reproductive processes in other mammals or in female patients or healthy volunteers after clinical translation.

The one point to be cleared in the present experiment is that whether the animals which were used in the study were observed in terms of regular estrus cycle before they began the experiment. In the manuscript, there is no any explanation about this important point. The authors have to mention that the animals used were chosen among those that show regular estrus cycle.

6. PLOS authors have the option to publish the peer review history of their article (what does this mean?). If published, this will include your full peer review and any attached files.

Reviewer #1: No

Reviewer #2: No

Reviewer #3: No

---

## [Author Response · Author response to Decision Letter 0]

6 Jul 2024

The authors thank the Editor and the Reviewers for their useful questions and remarks which have contributed considerably to improving the quality of our manuscript. Our answers are given below.

RESPONSE TO EDITOR

We are thankful for this remark; we have made all the necessary changes according to the requirements.

2. Thank you for stating the following financial disclosure. Please state what role the funders took in the study. Please include this amended Role of Funder statement in your cover letter; we will change the online submission form on your behalf.

Based on the request of the Editor, the following sentence is inserted in the manuscript: ‘The funders had no role in study design, data collection and analysis, decision to publish, or preparation of the manuscript.’

3. In the online submission form, you indicated that [Data are available for request]. All PLOS journals now require all data underlying the findings described in their manuscript to be freely available to other researchers, either 1. In a public repository, 2. Within the manuscript itself, or 3. Uploaded as supplementary information.

The data based on the research results are uploaded in table form as supplementary datasets.

The reference list has been checked and does not contain retracted papers. However, based on Reviewers' requests, the complete list has been supplemented; we have added 3 additional literature references to the complete list, which are highlighted in the manuscript.

RESPONSE TO REVIEWERS

REVIEWER 1

1) It would be better to give a brief and appropriate explanation about Cetrorelix and its pharmacological mechanism of action in the Introduction section.

The authors thank the Reviewer for this remark. The following has been inserted in the Introduction: ‘The physiological processes induced by the increase in the LH level can be inhibited by drugs. Cetrorelix is a potent and selective competitive antagonist of the GnRH receptor. Reversibly reduces LH and FSH from the anterior pituitary gland, thus preventing ovulation and inhibiting the production of sex hormones E2 and P4 sex hormones [5]. ‘

2) In the Methods section, which variance analysis (ANOVA, etc) was performed for multiple comparisons should be stated in the Statistical analysis sub-section.

In our experiments, we compared the values of two different groups of rats, so after parametric analysis of the data, we conducted unpaired t-tests as statistical analyzes. Therefore, it was impossible to make multiple comparisons.

3) References should be added to the use of Jongh's solution for the uterus, Tyrode's solution for the cecum, and the contractility study method.

New references have been inserted into the text and the reference list based on this comment.

(Gaddum JH. THE TECHNIQUE OF SUPERFUSION. Br J Pharmacol 1997;120(Suppl 1):82. 

Szűcs et al. Detection of stress and the effects of central nervous system depressants by gastrointestinal smooth muscle electromyography in wakeful rats. Life Sci. 2018;205:1–8.)

4) For the Figure 1: The figure should be adjusted correctly, as if KCl application on uterine contractions begins at the 15th minute rather than the 16th minute.

We apologize for this mistake. The time axis indicating the 30-minute time frame has been modified in Figure 1.

REVIEWER 2

1. What is the basis for the dose and duration of cetrorelix acetate treatment?

The dose and duration of cetrorelix treatment were established based on data from the previously published literature. Among the treatment protocols followed by the various working groups, the most frequently used dose of 0.5 mg/bwkg was administered over a 7-day period. New references have been inserted into the text and the reference list based on this comment.

2. What practical indications are that de Joung solution was used for uterus samples and Tyrode buffer for cecum samples in isolated organ baths?

Based on several decades of laboratory practice, we use two different buffer solutions during the examination of uterus and cecum samples. I have also confirmed this in the manuscript with accepted international publications. New references have been inserted into the text and the reference list based on this comment.

(Gaddum JH. THE TECHNIQUE OF SUPERFUSION. Br J Pharmacol 1997;120(Suppl 1):82. 

Szűcs et al. Detection of stress and the effects of central nervous system depressants by gastrointestinal smooth muscle electromyography in wakeful rats. Life Sci. 2018;205:1–8.)

3. Line 153 of the results section lacks a precise description of the in vivo smooth muscle electromyography result. Supplement the legend of Figure 3 with the Fourier transform FFT notation!

We are thankful for this remark, the Reviewer is right; however, line 153 refers to the interpretative diagram shown in Figure 3, the actual changes between the animal groups are described in Figure 4 and lines 159-160. On the other hand, the legend of Figure 3 was supplemented with missing information about the FFT abbreviation.

4. I think the bibliographic reference in line 158 (the legend of Figure 3) is redundant. Literature 11 can be found in the methods section and the summary also.

Thank you for your valuable comment; the mentioned reference has been removed from the legend of Figure 3.

5. In line 175: "Cetrorelix suppressed staining in the uterus (bottom image row)." I suggest instead, for the reader's ease of understanding, "Cetrorelix treatment reduced αvβ3 integrin expression, which caused a significant decrease in intensity."

Based on the Reviewer's remark, the legend of Figure 5 has been changed as suggested.

6. In Figure 5, I propose to mark the uterine horns with a white arrow in each image.

On the basis of this observation, we have modified Figure 5. 

7. Why the effect on KCl-induced contractions was considered to be justified in addition to the spontaneous uterine contractility study.

In isolated organ contractility studies, the examination of spontaneous contractions can be considered physiological; however, it is necessary to monitor the increase in contraction by a stimulus. This stimulus can be triggered with electrical voltage, a drug or - which we used in our study - KCl solution. From the degree of response to the stimulus, we can draw conclusions about the degree of structural or functional change caused by different treatment protocols. Furthermore, it should be noted that although the uterus is characterized by regular spontaneous contraction (which increases during pregnancy), this cannot be said for the cecum or other gastrointestinal sections. Therefore, KCl-stimulated contractions provide equal circumstances for the comparison of changes in the two different types of smooth muscle.

REVIEWER 3

The one point to be cleared in the present experiment is that whether the animals which were used in the study were observed in terms of regular estrus cycle before they began the experiment. In the manuscript, there is no any explanation about this important point. The authors have to mention that the animals used were chosen among those that show regular estrus cycle.

We are grateful to the Reviewer for this remark; we have detailed the relevant part in the methodology. The following section has been changed in the manuscript: ‘The estrus cycle of healthy young adult female rats was followed and measured daily using the Estrus Cycle Monitor (IM-01, MSB-MET Ltd., Balatonfüred, Hungary). Rats with vaginal impedance values of 4.5 to 7.5 kΩ were considered to be in the proestrus phase and were included in the experiment.’

---

## [Decision Letter · Decision Letter 1]

16 Jul 2024

Smooth muscle electromyography for detecting major alterations in the estrus cycle in rats

PONE-D-24-11044R1

Dear Dr. Gaspar,

We’re pleased to inform you that your manuscript has been judged scientifically suitable for publication and will be formally accepted for publication once it meets all outstanding technical requirements.

Kind regards,

Abeer El Wakil, PhD

Academic Editor

PLOS ONE

Additional Editor Comments (optional):

Reviewers' comments:

Reviewer's Responses to Questions

**Comments to the Author**

1. If the authors have adequately addressed your comments raised in a previous round of review and you feel that this manuscript is now acceptable for publication, you may indicate that here to bypass the “Comments to the Author” section, enter your conflict of interest statement in the “Confidential to Editor” section, and submit your "Accept" recommendation.

Reviewer #1: All comments have been addressed

Reviewer #2: All comments have been addressed

2. Is the manuscript technically sound, and do the data support the conclusions?

Reviewer #1: Yes

Reviewer #2: Yes

3. Has the statistical analysis been performed appropriately and rigorously? 

Reviewer #1: Yes

Reviewer #2: Yes

4. Have the authors made all data underlying the findings in their manuscript fully available?

Reviewer #1: No

Reviewer #2: Yes

5. Is the manuscript presented in an intelligible fashion and written in standard English?

Reviewer #1: Yes

Reviewer #2: Yes

6. Review Comments to the Author

Reviewer #1: The authors have redacted the minor revisions mentioned; therefore the manuscript is suitable for the publication.

Reviewer #2: I accept the answers of the authors and the corrections in the text. I recommend the manuscript for publication.

7. PLOS authors have the option to publish the peer review history of their article (what does this mean?). If published, this will include your full peer review and any attached files.

Reviewer #1: No

Reviewer #2: No

---

## [Editor Report · Acceptance letter]

29 Jul 2024

PONE-D-24-11044R1 

PLOS ONE

Dear Dr. Gaspar, 

I'm pleased to inform you that your manuscript has been deemed suitable for publication in PLOS ONE. Congratulations! Your manuscript is now being handed over to our production team.

Kind regards, 

on behalf of

Professor Abeer El Wakil 

Academic Editor

PLOS ONE